# Novel Study on Chemical Characterization and Antimicrobial, Antioxidant, and Anticholinesterase Activity of Essential Oil from Ecuadorian Bryophyte *Syzygiella rubricaulis* (Nees) Stephani

**DOI:** 10.3390/plants13070935

**Published:** 2024-03-23

**Authors:** Vladimir Morocho, Ángel Benitez, Bárbara Carrión, Luis Cartuche

**Affiliations:** 1Departamento de Química, Universidad Técnica Particular de Loja (UTPL), Loja 1101608, Ecuador; lecartuche@utpl.edu.ec; 2Biodiversidad de Ecosistemas Tropicales-BIETROP, Herbario HUTPL, Departamento de Ciencias Biológicas y Agropecuarias, Universidad Técnica Particular de Loja (UTPL), San Cayetano s/n, Loja 1101608, Ecuador; arbenitez@utpl.edu.ec; 3Carrera de Bioquímica y Farmacia, Universidad Técnica Particular de Loja (UTPL), Loja 1101608, Ecuador; becarrion1@utpl.edu.ec

**Keywords:** *S. rubricaulis*, essential oil, acetylcholinesterase, MIC, scavenging capacity, bicyclogermacrene, viridiflorol

## Abstract

Our research focuses on exploring the chemical composition and some biological properties of the essential oil derived from *Syzygiella rubricaulis* (Nees) Stephani, a bryophyte species. To conduct a comprehensive analysis, we utilized a DB5MS capillary column along with gas chromatography coupled to mass spectrometry (GC-MS) and flame ionization (GC-FID). The qualitative and quantitative examination revealed the presence of 50 compounds, with hydrocarbon sesquiterpenes (48.35%) and oxygenated sesquiterpenes (46.89%) being the predominant constituents. Noteworthy compounds identified include bicyclogermacrene (12.004%), cedranone <5-> (9.034%), spathulenol (6.835%), viridiflorol (6.334%), silphiperfol-5,7(14)-diene (6.216%), biotol <β-> (6.075%), guaiol (4.607%), viridiflorene (4.65%), and α-guaienol (3.883%). Furthermore, we assessed the antimicrobial, antioxidant, and anticholinesterase activity of the essential oil, revealing a compelling inhibitory effect against acetylcholinesterase (AChE) with an IC_50_ value of 26.75 ± 1.03 µg/mL and a moderate antimicrobial (MIC 500 µg/mL, *Enterococcus faecium*, *Lysteria monocytogenes*) and antioxidant effect (ABTS: SC_50_ 343.38 and DPPH 2650.23 µg/mL). These findings suggest the potential therapeutic application of the bryophyte essential oil in the treatment of Alzheimer’s disease due to its potent anticholinesterase properties.

## 1. Introduction

Bryophytes, believed to have evolved from freshwater algae, represent some of the earliest terrestrial organisms. They are characterized by their lack of vascular tissue and typically thrive in moist environments, reflecting their Greek-derived name meaning “plants that swell with water” [1]. Mosses, liverworts, and anthocerophytes are among the prominent groups of bryophytes, each playing a significant role in terrestrial ecosystems [2].

Despite their ecological importance, bryophytes have historically received less attention in scientific research compared to other plant groups. This oversight has resulted in a dearth of knowledge concerning the phytochemistry and biological activities associated with bryophytes [3]. Several factors contribute to this knowledge gap, including the challenges associated with obtaining sufficient plant material for analysis. Bryophytes are often small in size and tend to grow alongside other plant species, making it difficult to isolate and study them in isolation [4].

In essence, while bryophytes are recognized for their ecological significance, their study has been hindered by logistical and methodological challenges. Nonetheless, understanding their chemical composition and biological activities could unveil promising insights into their potential applications in various fields, from ecology to medicine. Efforts to overcome these challenges and explore the untapped potential of bryophytes continue to be essential for advancing scientific knowledge and conservation efforts.

Bryophytes are primarily found in humid environments where they form distinctive spheres and cushions on diverse substrates such as the ground, rocks, tree trunks, and even the leaves of vascular plants [3]. Despite their relatively simple morphological structure, bryophytes display a remarkable degree of chemical diversity [5]. In China, mosses have been extensively utilized for medicinal purposes. These plants have been employed in traditional medicine to address a variety of health concerns, including the treatment of burns, bruises, external wounds, snake bites, pulmonary tuberculosis, asthenia, fractures, convulsions, scalds, uropathy, pneumonia, and more [4].

Liverworts exhibit the capacity to produce a diverse range of secondary metabolites, including lipophilic terpenoids, acetogenins, and aromatic compounds (Asakawa, 2014). Notably, bryophytes, particularly mosses and liverworts, have been identified as possessing significant chemical and biological properties [6]. These plants serve as valuable reservoirs of biologically active compounds, and studies have underscored their potential in various domains. For instance, they have demonstrated notable antifungal properties [7], along with antibacterial and antiviral effects [8]. Furthermore, both mosses and liverworts showcase anti-inflammatory and antioxidant potential, and they have displayed promising anticancer activity in various cancer cell lines [9]. Leveraging these properties, bryophytes have been traditionally employed in India for treating wounds, cuts, burns, and skin diseases [10].

In the analysis of essential oils (EOs) from various bryophyte species, significant chemical components have been identified. For instance, *Breutelia tomentosa* exhibits epizonarene (8.7%) and α-selinene (6.7%), while *Leptodontius viticulosoides* contains β-selinene (13.5%) and α-selinene (10.5%). *Macromitrium perreflexum* is characterized by selina-3,11-dien-6-α-ol (19.7%) and curcuphenol (10.6%), and *Campylopus richardii* features epi-α-muurulol (15.1%) and α-cadinol (12.5%). *Rhacocarpus purpurascens* shows α-cadinol (36.8%) and α-santalene (8.4%), and *Thuidium peruvianum* includes phytol (21.7%) and valerenol (10.1%). These findings shed light on the diverse chemical profiles of bryophyte essential oils, hinting at the potential presence of bioactive compounds with applications in various fields, including medicine and industry [11].

*Syzygiella rubricaulis* (Nees) Stephani (Figure 1), classified as a bryophyte [12], is a dioecious leafy liverwort native to Neotropical mountains and the Azores, typically thriving above 1000 m. This liverwort forms crusts on exposed rocks or rocky outcrops and features erect shoots with few branches and a decurved apex. With a height ranging from 2 to 8 cm, the plant exhibits a yellowish-green color that varies from brownish-orange to reddish-brown. The laterally ovate–obicular leaves have a rounded apex, large trigonous cells, and sparse rizoidal cuticles [13]. Previous research on *S*. *rubricaulis* has identified notable chemical constituents in the extract, including significant compounds such as theaspirane, recognized for its aromatic properties in various tea types, the diterpene fusicoccan-2,5-diene, and peculiaroxide [14].

This starting study on the essential oils of *Syzygiella rubricaulis* from Ecuador highlights its pioneering nature, exploring the chemical composition and acetylcholinesterase (AChE) activity, as well as the antimicrobial and antioxidant effects. The research represents a significant step in advancing the understanding of essential oils in bryophytes. However, it is important to note that the investigation is still in its early stages, and further studies and validations are needed before drawing conclusive implications for future research development in the pharmaceutical industry. Beyond its scientific significance, the study contributes to enhancing the effectiveness and quality of Ecuadorian species, promoting sustainable development, and valorizing traditional preparations based on local natural resources.

## 2. Results

### 2.1. GC/MS Analysis of Essential Oil

Gas chromatography–mass spectrometry (GC-MS) analysis of the essential oils (EOs) extracted from *Syzygiella rubricaulis* unveiled the presence of 50 compounds, collectively constituting 95.63% of the total essential oil composition (refer to Table 1 and Figure 2 for details). Noteworthy compounds include Bicyclogermacrene (12.004%), 5-Cedranone (9.034%), Spathulenol (6.835%), Viridiflorol (6.334%), Silphiperfol-5,7(14)-diene (6.216%), β-Biotol (6.075%), Guaiol (4.607%), Viridiflorene (4.65%), and α-Guaiene (3.883%). Hydrocarbon sesquiterpenes (48.35%) and oxygenated sesquiterpenes (46.89%) collectively dominate the characteristic groups of these compounds. This detailed analysis provides valuable insights into the chemical composition of the essential oils from *Syzygiella rubricaulis*, laying the foundation for further exploration of its potential applications in various fields.

### 2.2. Antimicrobial Activity of Essential Oil

The broth microdilution method was employed to assess the antimicrobial activity of the essential oil. *S*. *rubricaulis* EO demonstrated moderate inhibitory potential against *Enterococcus faecium* ATCC^®^ 27,270 and *Listeria monocytogenes* ATCC^®^ 19,115, with an MIC value of 500 µg/mL. Other tested microorganisms displayed weak or negligible inhibitory potential at concentrations exceeding 1000 µg/mL. Positive controls included ampicillin, ciprofloxacin, and amphotericin B, while dimethyl sulfoxide served as the negative control. The results of the antimicrobial activity are presented in Table 2.

### 2.3. Scavenging Radical Capacity of Essential Oil

The antioxidant capacity of the EO of *S*. *rubricaulis* exhibited moderate to weak potential, with an SC_50_ value of 343.38 ± 0.41 and 2650.23 ± 25.42 µg/mL in ABTS and DPPH assays, respectively. Trolox equivalent antioxidant capacity (TEAC) was also calculated from ABTS data and expressed as µM Trolox/g EO. Trolox was also used as a positive reference control (Table 3).

### 2.4. AChE Analysis of Essential Oil

The anticholinesterase activity of the essential oil (EO) from *Syzygiella rubricaulis* was assessed for the first time in this study. The oil demonstrated notable activity against acetylcholinesterase (AChE), with an IC_50_ value of 26.75 ± 1.03 μg/mL, as illustrated in Figure 3. In comparison, the synthetic drug, donepezil hydrochloride, commonly used for treating mild to moderate forms of Alzheimer’s dementia, served as a positive control in the test and exhibited an IC_50_ value of 12.40 ± 1.35 nM against AChE [16]. These findings underscore the potential of *S. rubricaulis* EO as a source of compounds with anticholinesterase activity, suggesting its potential relevance in the context of Alzheimer’s disease research and treatment.

## 3. Discussion

Using a DB5MS capillary column, the essential oil extracted from *Syzygiella rubricaulis* was analyzed, revealing a substantial presence of hydrocarbon sesquiterpenes (48.352%) and oxygenated sesquiterpenes (46.885%). The major compounds identified include bicyclogermacrene (12.004%), 5-cedranone (9.034%), spathulenol (6.835%), viridiflorol (6.334%), silphiperfol-5,7(14)-diene (6.216%), *β*-biotol (6.075%), guaiol (4.607%), viridiflorene (4.65%), and *α*-guaiene (3.883%).

The phytochemical results obtained in this study cannot be directly compared with existing literature data due to the absence of prior phytochemical investigations on *Syzygiella rubricaulis*. Limited scientific publications exist on phytochemical studies of liverwort, with one notable example being the research conducted by Valarezo et al. (2020). In their study on bryophytes, hydrocarbon sesquiterpenes (65.39%) and oxygenated sesquiterpenes (23.48%) were identified using the DB5-MS column. Key compounds with the highest proportion included bicyclogermacrene (8.42%), viridiflorene (6.51%), and cis-thujopsene (7.00%), all belonging to the category of hydrocarbon sesquiterpenes. These findings align with our data, underscoring the richness of these compounds in *S. rubricaulis* and providing valuable confirmation [9]. In the case of *S*. *anomala*, analysis using DB5-MS revealed the presence of compounds such as silphiperfola-5,7(14)-diene (25.22%), cis-thujopsene (7.00%), viridiflorene (6.51%), β-vetispirene (8.01%), bicyclogermacrene (8.42%), caryophyllene oxide (8.98%), and β-Oplopenone (6.40%). While these compounds are present in varying proportions, they share characteristics with those identified in our study. This suggests a certain consistency in the presence of these compounds in bryophytes, highlighting their richness in such chemical constituents [17]. In the study of another species, *Syzygiella anomala*, it was demonstrated that sesquiterpenes constituted the predominant group of compounds. Notable among these were silphiperfola-5,7(14)-diene (25.22%), caryophyllene oxide (8.98%), bicyclogermacrene (8.42%), and β-vetispirene (8.01%) [9].

Likewise, in another investigation focusing on Ecuadorian liverwort, specific compounds such as 3,4-Dimethoxy-1-vinylbenzene, 2,4,5-trimethoxy-1-vinylbenzene, and apigenin-7,4′-dimethylether were identified in *Marchesinia brachiata* [18]. Additionally, arbusculina B and (-)-α-bisabolol were found in *Frullania brasiliensis*, while 1,4-Dimethylazulene was isolated from *Plagiochila micropterys* and *Macrolejeunea pallescens* [19].

In Costa et al.’s (2018) research, spanning Ecuador, Mexico, Venezuela, and Brazil, an examination of *S*. *rubricaulis* extracts revealed the identification of approximately 50 compounds. The proportions of these compounds were found to be influenced by various factors, such as increased radiation and low temperatures at higher altitudes, suggesting that these conditions impact the secondary metabolism in plants. Additionally, less-explored factors like soil type, wind speed, snow cover, vegetation, and radiation were also recognized as potential contributors to the quantitative and qualitative determination of metabolites [14]. Furthermore, as noted by Turek and Stintzing (2013), the time of harvest or collection, as well as processes like oxidation, isomerization, cyclization, or dehydrogenation, can serve as additional determining factors in the composition of secondary metabolites [20].

In the study by Yang et al. (2018), it is elucidated that metabolites exhibit adaptability, generating viability in response to environmental stress. The research suggests that a warm climate can enhance the yield of sesquiterpenes in certain plant organisms. Similarly, the relative humidity in the planting area was identified as a direct influencer on metabolite concentrations, indicating that lower water stress or drought conditions may lead to the accumulation of higher concentrations of metabolites [21]

The diversity of terpenes in bryophytes is linked to the existence of oil bodies, which are unique and prominent organelles specific to these spore-forming plants. Many of these compounds boast identifiable structures, some of which are unprecedented in any other species. These terpenes play crucial roles in diverse biological and ecological processes, showcasing unique properties that make them intriguing from a biological standpoint [3,22].

Chen et al. (2018) observed in their analysis that a distinctive feature of bryophytes is the occurrence of sesquiterpenoid compounds, which accumulate in the characteristic oil bodies of these plants. The study further highlights that UV-B radiation exerts an impact on both physiological and chemical processes, leading to the degradation of proteins, nucleic acids, and other macromolecules. Terpenoids, in this context, can function as photo-receptors, influencing the regulation of secondary metabolites [23].

Despite its moderate antimicrobial potential exhibited by *S. rubricaulis* EO against two gram-positive bacteria, the study of the non-volatile fraction could be the next logical research step to not discard the pharmacological potential of this species, as revealed by Quenon et al. (2022), who conducted an antimicrobial study of a related species, *Sizygiella malaccense*, and identified seven new alkyl–salicylic acids with antimicrobial potential against *Sthapylococcus aureus*, with MIC values ranging from 18.75 to 75.0 µg/mL, *Streptococcus pyogenes* (2.34 < MIC < 18.75 µg/mL), and *Pseudomonas aeruginosa* with an MIC value of 150 µg/mL) [24]. To the best of our knowledge this is the first report of the antimicrobial potential of *S. rubricaulis*; however, GC/MS research conducted by Costa et al. (2018) revealed the occurrence of 50 different compounds in 12 different populations of *S. rubricaulis* (four located in Ecuador, two from Venezuela, one in México, and six in Brazil), with most of them belonging to the sesquiterpenes structural class, followed by hydrocarbons, but nothing is said about the biological properties, such as antimicrobial, antioxidant, or related activities [14].

Similar research has been undertaken in Ecuador regarding the chemical composition of *Sizygiella* species, such as *Syzygiella anomala* collected in the region. Analysis of the essential oil through GC/MS revealed the presence of 27 compounds, predominantly of sesquiterpene nature (>88% of the total oil composition). However, biological properties were not assessed in this study [9].

Although literature is scarce about the antimicrobial effect of *Sizygiella* spp. essential oil, we can compare the chemical composition of volatile fractions of liverworts like the research conducted by Bukvicki et al. (2012), who identified, through GCMS-SPME, the predominant occurrence of sesquiterpene hydrocarbons (23 to 53% of the chemical composition), followed by monoterpene hydrocarbons (ca. 22.83%), non-terpene hydrocarbons, and alcohols. The most representative compounds were β-phellandrene (15.54% β-caryophyllene (10.72%), neoisolongifolene (6.25%), ç-Gurjunene (5.15%), and p-Cymene (3.11%). Regarding the antimicrobial effect, all the extracts were tested against yeast, exerting a mild to weak effect (0.5 to 3 mg/mL, MIC), and against *L. monocytogenes,* displaying a weak effect, with MIC values from 1 to 2 mg/mL [25]. According to these results, the antimicrobial effect could be attributed to the highest content of sesquiterpene hydrocarbons. Likewise, in our research, the high content of sesquiterpenes could be responsible for the antimicrobial effect observed against *L. monocytogenes* and *E. faecium*.

In another related study of essential oil from bryophytes, it was found that the EO of *Antitrichia curtipendula* exhibited moderate antibacterial activity against gram-positive bacteria, like *S. aureus, E. faecalis*, and *S. mutans*, and also displayed a good antioxidant effect by DPPH and the chelating ion method with inhibition percentages of 18.11 and 67.42%, respectively. This antibacterial and antioxidant effect could be attributed to the high content of biformene in the EO (13.06%) [26].

Concerning antioxidant capacity, there are no studies about the scavenging capacity of essential oils from this species. However, bryophytes, in general, are considered a rich source of natural antioxidants. Numerous bryophytes have been documented to exhibit notable antioxidant activity. Some of them demonstrate highly efficient antioxidant enzyme systems, whereas others showcase a variety of phenolic and flavonoid compounds responsible for scavenging free radicals. Bryophytes have been conventionally utilized in Chinese, Indian, and American societies for diverse medicinal purposes. Nevertheless, the ethnomedicinal application of bryophytes requires thorough scientific investigation and validation [27].

Bryophythes have found commercial applications due to their biological activities, and their products are sold in Germany as fungicides in agriculture, antifeedant agents, and mosses, particularly as cover for roofs [28]. The antimicrobial effect is attributed to the ability to produce a large variety of chemical entities like polifenols and terpenes, but essential oils have not been deeply studied for the same applications.

In our investigation of essential oils possessing anticholinesterase properties, we made a novel discovery by demonstrating the selective inhibition of ChE enzymes by *S*. *rubricaulis* EOs. Specifically, the essential oil displayed an IC_50_ of 26.75 ± 1.03 μg/mL against the AChE enzyme. Notably, Gupta et al. (2001) conducted a study involving 39 bryophyte species from 16 families, revealing detectable ChE activity in 30 species from 13 families. Among these, *Anoectangium bicolor* of the Pottiaceae family exhibited the highest ChE activity (240 pmol ATCh hydrolyzed. s^−1^·g^−1^ fresh mass) using Ellman’s test for ChE activity [29]. Given that bryophytes are considered an excellent system for experimental studies in plant physiology and biochemistry, the identification of ChE in our study suggests its potential utility for exploring the involvement of the acetylcholine system in signal transduction pathways in plants.

Numerous studies have consistently demonstrated that essential oils with elevated concentrations of sesquiterpenes display robust and potent acetylcholinesterase (AChE) inhibitory activity [30]. Venditti et al. (2018) further supported this observation, noting that as the content of sesquiterpenoids increased relative to monoterpenoids, there was a corresponding enhancement in the essential oil’s ability to inhibit AChE [31].

One of the main components of the EO, aromadendrane viridiflorol (6.33% in our study), is known for its antibacterial effect, particularly against *M. tuberculosis,* and its anti-inflammatory and antioxidant activity [32]. It has been demonstrated by Gilabert et al. (2015) that viridiflorol, a sesquiterpene isolated from the liverwort *Lepidozia chordulifera*, was able to inhibit bacterial growth at a moderate rate and prevent biofilm formation in *S. aureus* (40%) and *P. aeruginosa* (60%) at 50 µg/mL [33]. Additionally, spathulenol (found at 6.83% in our study) could be responsible for the moderate antimicrobial effect observed, as reported by Fernandes et al. (2015), who identified a moderate antibacterial effect of spathulenol at 200 µg/mL against *Cryptococcus neoformans* and *E. faecalis* [34].

Much research remains necessary to authenticate the pharmacological potential of bryophytes. This endeavor entails initial scrutiny into the chemical heterogeneity of essential oils sourced from various species and geographic locales to pinpoint chemotaxonomic markers. Subsequently, biological assessments across a spectrum of in vitro and in vivo assays are indispensable. Our ongoing work serves as a foundational framework for further delving into the chemical diversity and functional attributes of bryophytes indigenous to Ecuador.

## 4. Materials and Methods

### 4.1. Sample Collection

Samples of *S. rubricaulis* were collected in the El Tiro sector, located at the border between Loja and Zamora Chinchipe, Ecuador, precisely at coordinates 3°59′08.8″ S 79°08′32.9″ W. The collection of plant material was carried out under the permit MAE-DBN-2016-048 issued by the Ministerio del Ambiente, Agua y Transición Ecológica (MAATE) of the Ecuadorian Government. Authentication of the plant material was performed by Angel Benitez, a botanist at the Herbarium UTPL. A specimen sample has been deposited at the Herbarium of the Universidad Técnica Particular de Loja (HUTPL) with the voucher code PPN-as-057.

### 4.2. Essential Oil Isolation

The fresh bryophyte was subjected to steam distillation using a Clevenger apparatus for a duration of 4 h, with three replicates of approximately 200 g each. The resulting three distillates were combined to obtain the pale-yellow essential oil (EO). Subsequently, the EO was dried using anhydrous Na_2_SO_4_ and stored in an amber vial at −7 °C until further biological and chemical analyses.

### 4.3. Chemical Characterization of Essential Oil

#### 4.3.1. Sample Preparation

Quantitative and qualitative characterization of the EO from *S*. *rubricaulis* required sample preparation of the volatile fractions. The EO was diluted (10 µL) with dichloromethane HPLC grade from Sigma (St. Louis, MO, USA) (990 µL). This procedure was conducted in triplicate, and the samples were used in the chemical analyses described below.

#### 4.3.2. Qualitative and Quantitative Analyses

For qualitative identification, gas chromatography coupled with mass spectrometry (GC-MS) was employed using a Thermo Fisher Scientific model Trace 1310 Gas Chromatograph (GC) equipped with Thermo Scientific AI/AS 1300 liquid sampling automation and model ISQ7000 Mass Spectrometer Single Quadrupole (Waltham, MA, USA). Experimental conditions involved electron impact mass spectra taken at 70 eV, with a mass range set at 40 to 400 m/z. Ultra-pure helium (from Indura, Guayaquil, Ecuador) served as the carrier gas at a constant flow of 1.00 mL/min. The oven temperature program was initiated at 60 °C for 5 min, followed by an increase to 250 °C at a 3 °C/min gradient operation. The ion source temperature was set at 230 °C, and the quadrupole temperature was set at 150 °C. The capillary columns used were DB-5 ms (5% phenylmethylpolysiloxane, 30 m × 0.25 mm id, 0.25 μm film thickness, J & W Scientific, Folsom, CA, USA). Compound identification was accomplished by comparing mass spectra and linear retention indexes (LRIs) with literature references. The LRI was experimentally determined following the method of Van Den Dool and Krats [35], involving the injection of a series of homologous C9 to C24 n-alkanes (ChemService, West Chester, PA, USA).

Quantitative analysis of EO was performed using gas chromatography coupled with a flame ionization detector (GC-FID). The prepared samples were injected using the same column under the same analytical conditions as the qualitative GC-MS method. Aromatic content was determined by comparing the total area of GC peaks with the identified peaks without the use of a corrector factor [36].

### 4.4. Broth Microdilution Assay

Antimicrobial potential was assessed following the protocol outlined by Cartuche et al. [37]. Minimum inhibitory concentration and antimicrobial potential were determined using the broth microdilution technique. A set of American Type Culture Collection (ATCC) strains was chosen for the assay. These strains are characterized as opportunistic human microorganisms responsible for common human infectious diseases and are widely employed worldwide for antimicrobial tests. Three cocci bacteria (*E. faecalis* ATCC^®^ 19433, *E. faecium* ATCC^®^ 27270, *S. aureus* ATCC^®^ 25923), four rod-shaped bacteria (*L. monocytogenes* ATCC ^®^ 19115, *E. coli* (O157:H7) ATCC^®^ 43888, *P. aeruginosa* ATCC^®^ 10145, *S. enterica* ATCC ^®^14028), one microaerophile rod-shaped bacteria (*C. jejuni* ATCC ^®^ 33560), and two fungi (*C. albicans* ATCC^®^ 10231, *A. niger* ATCC^®^ 6275) were selected as microbial test models. Concentrations ranging from 4000 to 31.25 µg/mL were achieved using the double serial dilution method, with a final inoculum concentration of 5 × 10^5^ cfu/mL for bacteria, 2.5 × 10^5^ cfu/mL for yeasts, and 5 × 10^4^ spores/mL for sporulated fungi. Mueller Hinton II (MH II) and Sabouraud broths were utilized as test media for bacteria and fungi, respectively. The *Campylobacter jejuni* culture was activated by adding horse serum at 5% in Tioglycolate medium and incubated under microaerophilic conditions (Campygen sachet, Thermo Scientific) at 37 °C for 48 h. The broth microdilution test was performed in MH II supplemented with 5% horse serum (Thermo) under the same conditions as described previously.

Commercial antimicrobial agents served as positive controls: ampicillin solution (1 mg/mL) for *E. faecalis*, *E. faecium*, and *S. aureus*; ciprofloxacin solution (1 mg/mL) for *P. aeruginosa*, *S. enterica*, and *E. coli*; Erithromycin solution (1 mg/mL) for *C. jejuni*; and amphotericin B solution (250 µg/mL) for *C. albicans* and *A. niger*.

### 4.5. Radical Scavenging Capacity

#### 4.5.1. 2,2-Diphenyl-1-picrylhydrazyl Radical Scavenging Assay

The DPPH radical scavenging assay was carried out in accordance with the procedure proposed by Cartuche et al. [37], utilizing the 2,2-diphenyl-1-picrylhydrazyl (DPPH) free radical. A working solution was prepared by dissolving 24 mg of DPPH in 100 mL of methanol and stabilized in an EPOCH 2 microplate reader (BIOTEK, Winooski, VT, USA) at 515 nm until an absorbance of 1.1 ± 0.01 was attained. The antiradical interaction between the essential oil (EO) and free radicals was conducted at various EO concentrations (1, 0.5, and 0.25 mg/mL). Subsequently, 270 µL of the DPPH-adjusted working solution and 30 µL of the EO sample were placed in a 96-microwell plate. The reaction was monitored at 515 nm for 60 min at room temperature. Trolox and methanol were employed as the positive control and blank control, respectively. The outcomes were expressed as SC_50_ (sweep concentration of the radical at 50%). All measurements were performed in triplicate to ensure accuracy.

#### 4.5.2. 2,2-Diphenyl-1-picrylhydrazyl Radical Scavenging Assay

The antioxidant power measured using the ABTS cation radical was conducted under the guidelines provided by Cartuche et al. [37]. In summary, the assay commenced with the creation of a radical stock solution by combining equal volumes of ABTS (7.4 µM) and potassium persulfate (2.6 µM), followed by 12 h of stirring. The standard solution was then prepared by dissolving it in methanol to achieve an absorbance of 1.1 ± 0.02, measured at 734 nm using an EPOCH 2 microplate reader (BIOTEK, Winooski, VT, USA). The antiradical reaction was assessed for 1 h in the dark at room temperature, involving the addition of 270 µL of the adjusted ABTS working solution and 30 µL of the M. discolor EO at various concentrations (1, 0.5, and 0.25 mg/mL) into a well. Trolox and methanol served as the positive and blank controls, respectively. The outcomes were reported as SC50 (sweep concentration of the radical at 50%).

### 4.6. Cholinesterase Assay

The in vitro assessment of acetylcholinesterase inhibitory activity for *S*. *rubricaulis* essential oil (EO) followed the method outlined by Ellman et al. [38] with slight modifications, as proposed by Andrade et al. in 2022 [16]. In brief, the reaction mixture consisted of 40 μL of Tris Buffer, 20 μL of the analyzed sample solution, 20 μL of acetylthiocholine (ATCh, 15 mM PBS, pH 7.4), and 100 μL of DTNB (3 Mm, Tris Buffer). A preincubation period of 3 min at 25 °C with continuous shaking was followed by the addition of 20 μL of 0.5 U/mL AchE to initiate the reaction. The released product was monitored at 405 nm on an EPOCH 2 microplate reader (BIOTEK 1) at 25 °C for 60 min.

The essential oil (EO) of *S*. *rubricaulis* was prepared by dissolving 10 mg in 1 mL of MeOH. Four additional dilutions (10× factor dilution) were made to achieve final concentrations of 1000, 500, 100, 50, and 10 μg/mL. Progression curves were generated from the absorbance using a standard curve of DTNB and L-GSH at various molar concentrations to measure the rate, expressed as mM/min of product released. MeOH, chosen as a non-selective protic solvent, served as a negative control at a maximum concentration of 10% in the final volume without impacting the enzymatic reaction. Donepezil hydrochloride was employed as a positive control, exhibiting a calculated IC_50_ value of 12.40 ± 1.35 nM.

## 5. Conclusions

For the first time, the chemical composition, physical characteristics, and biological properties of *S. rubricaulis* essential oil (EO) were thoroughly investigated and documented. Hydrocarbon sesquiterpenes were identified as the predominant components, constituting around 50% of the oil composition. The EO demonstrated dose-dependent inhibition of cholinesterase (ChE) activity, particularly exhibiting notable selectivity towards acetylcholinesterase (AchE). These initial findings set the foundation for future research endeavors aimed at identifying the specific oil components responsible for the inhibitory effect and elucidating the kinetic mechanisms underlying their interaction with the cholinesterase enzyme. The overarching goal is to explore the potential integration of *S. rubricaulis* EO into natural therapeutic approaches for the treatment of Alzheimer’s disease and other related dementias. Despite the essential oil’s moderate to low antimicrobial and antioxidant efficacy, this study represents the inaugural report for this species and opens avenues for further exploration of potential medicinal applications of bryophytes.

## Figures and Tables

**Figure 1 plants-13-00935-f001:**
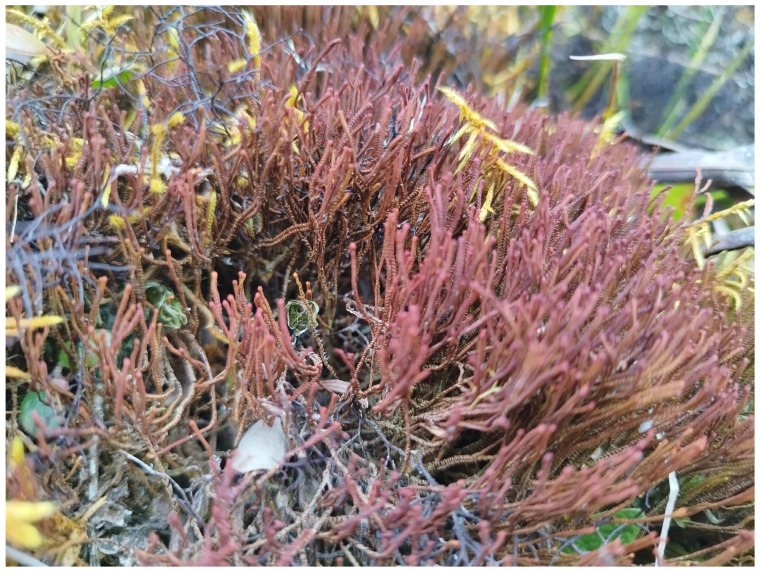
*Syzygiella rubricaulis* (Nees) Sthephi. collected in Ecuador. Photo provided by one of the authors (A.B.).

**Figure 2 plants-13-00935-f002:**
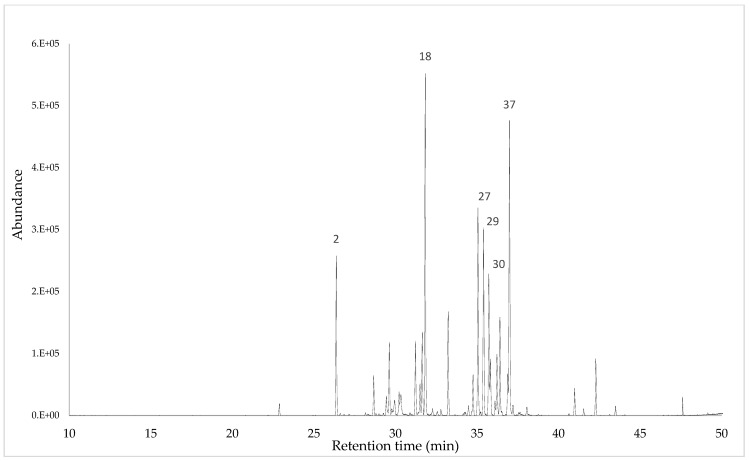
Typical GC-MS chromatogram of essential oil obtained with a DB5-MS capillary column. The numbers correspond to the main components.

**Figure 3 plants-13-00935-f003:**
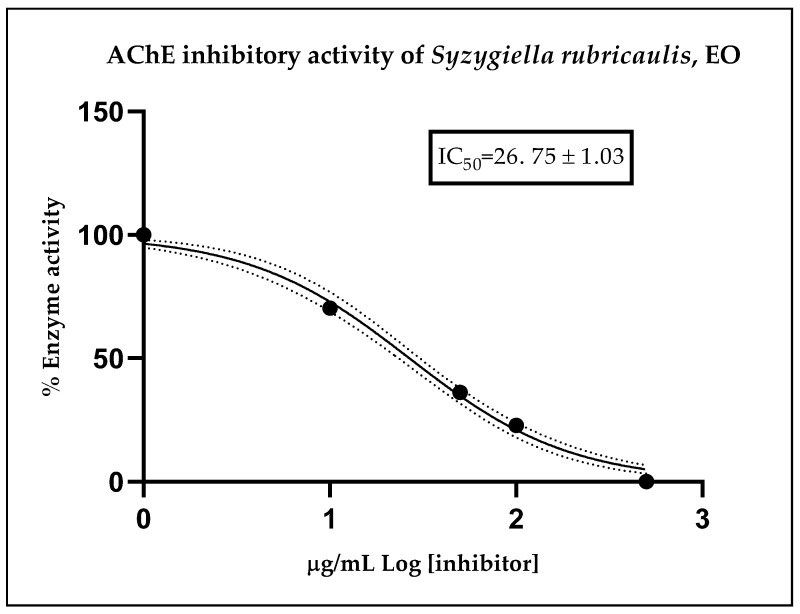
Dose–response curve of cholinesterase residual activity (%) vs. concentration (μg/mL) of the EO from *S. rubricaulis*, used to calculate the IC_50_ values against the AChE enzyme.

**Table 1 plants-13-00935-t001:** Chemical composition of *Syzygiella rubricaulis* (Nees) Sthephi. Essential oil obtained with DB5-MS non-polar capillary columns.

N.	Compounds	DB5-MS		
		LRIcal ^a^	LRIref ^b^	% ^c^
1	Bornyl acetate	1283	1284	0.436
2	Silphiperfol-5,7(14)-diene	1362	1358	6.216
3	α-Copaene	1367	1374	2.002
4	α-Gurjunene	1403	1409	0.241
5	*E*-Caryophyllene	1415	1417	2.127
6	*β*-Copaene	1429	1430	0.398
7	cis-Thujopsene	1434	1429	2.762
8	α-Guaiene	1438	1437	3.883
9	Myltayl-4(12)-ene	1442	1445	1.282
10	α-neo-Clovene	1446	1452	1.13
11	Sesquisabinene	1452	1457	1.809
12	cis-Cadina-1(6),4-diene	1455	1461	2.273
13	*β*-Acoradiene	1469	1469	0.27
14	γ-Muurolene	1477	1478	2.65
15	α-curcumene	1481	1479	0.135
16	cis-*β*-Guaiene	1483	1492	1.827
17	Viridiflorene	1487	1496	4.65
18	Bicyclogermacrene	1491	1500	12.004
19	δ-Selinene	1502	1492	0.598
20	γ-Cadinene	1510	1513	0.496
21	δ-Amorphene	1515	1511	0.384
22	Macrocarpene	1526	1526	3.948
23	Germacrene B	1550	1559	0.199
24	Selina-3,7(11)-diene	1552	1545	0.395
25	cis-Muurol-5-en-4-β-ol	1558	1550	0.729
26	Longipinanol	1565	1567	1.697
27	Spathulenol	1573	1577	6.835
28	*α*-Cedrene epoxide	1577	1574	0.31
29	Viridiflorol	1582	1592	6.334
30	Guaiol	1590	1600	4.607
31	cis-*β*-Elemenone (impure)	1592	1589	1.841
32	Globulol	1599	1590	0.54
33	Rosifoliol	1603	1600	2.454
34	*β*-Biotol	1607	1612	6.075
35	1,3,5-Bisabolatrien-7-ol	1610	1601	0.149
36	Isolongifolan-7-α-ol	1620	1618	1.91
37	5-Cedranone	1623	1628	9.034
38	allo-Aromadendrene epoxid	1629	1638	0.638
39	α-Cadinol	1638	1632	0.224
40	epi-*α*-Cadinol	1641	1638	0.5
41	7-epi-*α*-Eudesmol	1652	1662	0.124
42	Cryptomerione	1732	1724	1.417
43	Xanthorrhizol	1747	1751	0.396
44	14-hydroxy-*α*-Muurolene	1769	1779	0.477
	Hydrocarbon sesquiterpenes			48.35
	Oxygenated sesquiterpenes			46.89
	Others			0.44
	Total identified			95.64

^a^ Calculated linear retention index; ^b^ linear retention index from reference [15]; ^c^ mean percent content in the EO over three determinations.

**Table 2 plants-13-00935-t002:** Minimum inhibitory concentration calculated for *Syzygiella rubricaulis* EO against ten human–opportunistic microorganisms.

Microorganisms	*S. rubricaulis* Essential oil ^†^	Antimicrobial Agent (Positive Control) ^†^
**Cocci Bacteria**		**Ampicillin (1 mg/mL)**
*Enterococcus faecalis*	4000	0.7812
*Enterococcus faecium*	500	<0.3906
*Staphylococcus aureus*	4000	<0.3906
**Rod-shaped Bacteria**		**Ciprofloxacin (1 mg/mL)**
*Lysteria monocytogenes*	500	1.5625
*Escherichia coli* (O157:H7)	Non active	1.5625
*Pseudomonas aeruginosa*	Non active	<0.3906
*Salmonella enterica serovar Thypimurium*	4000	<0.3906
**Microaerophile Rod-shaped bacteria**		**Erithromycin (1 mg/mL)**
*Campylobacter jejuni*	Non active	15.625
**Yeasts and sporulated fungi**		**Amphotericin B (250 µg/mL)**
*Candida albicans*	4000	<0.098
*Aspergillus niger*	Non active	<0.098

† MIC values are given as µg/mL.

**Table 3 plants-13-00935-t003:** Half scavenging capacity (SC_50_) of *S. rubricaulis* EO.

EO	TEAC	ABTS	DPPH
*S. rubricaulis*	µM Trolox/g EO	SC_50_ ± SD (µg/mL—µM *)	
	96.74 ± 11.42	343.38 ± 0.41	2650.23 ± 25.42
Trolox *	-	24.72 ±1.03	28.97 ± 1.24

* Trolox was used as a positive reference and its values is given in µM.

## Data Availability

The data presented in this study are available on request from the corresponding author.

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
