# Peer review of "Novel Study on Chemical Characterization and Antimicrobial, Antioxidant, and Anticholinesterase Activity of Essential Oil from Ecuadorian Bryophyte Syzygiella rubricaulis (Nees) Stephani"

_plants, 2024, doi:10.3390/plants13070935_

Round 1

Reviewer 1 Report

Comments and Suggestions for Authors

The authors addressed an important issue related to the use of substances of natural origin. The use of Essential Oils is gaining more and more attention, so research in the context of identification of EOs commissions and potential methods of its use is of high significance. The general idea of this work remains unclear to me, hence my decision to reconsider this work after major revision.

The greatest value of this work is the very detailed and accurate identification of the chemical composition of Syzygiella rubricaulis (Nees) Sthephi belonging to bryophyte species. This issue is highlighted by authors in the discussion section. In lines 94-102, authors indicated that apart from exploring chemical composition of this EOs, the aim of this work was to investigate its antimicrobial and antioxidant activity. Moreover, authors indicated that “research represent crucial step in advancing the understanding of essential oils in mosses, potentially paving the way for the development of new pharmaceutical” and that “This research holds promise for both scientific advancements and the commercial utilization of bryophytes in Ecuador”. In my opinion, it is a positive thing that the performed research at least partially contributes to the creation of knowledge that can be commercially used. However, taking into consideration the results presented by the authors, this is too far-reaching vision of the use of these results. The vision is so distant that I consider it inappropriate to point it out in this scientific article.

Authors present results for: (1) MIC values for selected gram-positive and gram-negative bacteria as well as yeasts and sporulated fungi; (2) Scavenging radical capacity and(3) AChE Analysis. All these results are presented with a reference to commercially available products there were used as a positive control.

As for the results for MIC, it is shown that S.rubricaulis EOs have activity against the Gram-positive bacteria E. faecium ATCC® 27270 and L. monocytogenes ATCC® 19115 with a MIC value of 500 μg/mL. Authors indicated in line 131, that EO demonstrated its highest inhibitory potential against indicated microbes. It is an exaggerated statement, especially that there are no other EOs from different bryophyte species that were subjected to study. Based on presented results only, there is no commercial potential for this EOs, as the difference between commercial products and tested EOs is far too big. Presented results indicate that this EOs have such activity, but the MIC values cannot be compared to anything (meaning other EOs or other products different than used reference), and thus cannot be further discussed.  The same applies for other results presented in this paper.

As for the results for Scavenging radical capacity, there is a different value for DPPH assay provided in the text (line 142), which is equal to 2650.23 and in the table 3, which is equal to 1313.73.

The results obtained for EOs test  are closest to the reference value for the AChE study. IC50 value for this assay for EOs is equal to 26.75 μg/mL and the value for synthetic drug donepezil hydrochloride, commonly used for treating mild to moderate forms of Alzheimer's dementia is equal to 12.40 μg/mL.

However, in this case we rely only on numerical values and we do not know anything about other products or active substances showing such activity (both in terms of other EOs and other commercially available products). If the authors wanted to base their work on the value resulting from this result, it would be necessary to present the relationship between the tested activity and Alzheimer's disease.

The authors identified a given activities, but its identification alone is not enough for such far-reaching conclusions about a groundbreaking study (line 94). Especially since, as the authors point out (e.g. Lines 242-244, lines 230-235), it was suspected that this type of EOs (meaning EOs obtained from S. rubricaulis or from bryophytes species) would contain compounds with such activities. Thus, this result should be treated more like a confirmation rather than discovery of such activity. In my opinion, the work should be supplemented with the results for other EOs of S. rubricaulis or bryophytes origin. Different activities of tested EOs could then be correlated with different EOs composition. Such discussion would be of scientific value. And only after such research and discussion, it would be possible to come to the conclusion that given EOs has the highest activity among testes EOs, which provides grounds for conducting further research in this area. Following the authors way of thinking presented in this paper, each identification of a given activity in each of the EOs provide grounds for commercial application, which is not entirely true.

Current form of the paper make the discussion of the results pointless, which is the main drawback of this work. If the authors have any results for other oils, or will conduct additional analyzes within a reasonable time, I believe that the work will have the potential to be published in  Plants Journal.

Author Response

March, 2024

Prof. Dr. Dilantha Fernando

Editor in Chief

Plants

Subject: Online Manuscript Submission

To Editor

I am submitting a required revision for consideration in Plants journal with the changes made in the manuscript with the track-change mode, for all the suggestions made by reviewers as described below:

 REVIEWER 1

Thank you for every inquiry made, below you can find our responses to improve our work according to your suggestions.

Suggestion.

The authors addressed an important issue related to the use of substances of natural origin. The use of Essential Oils is gaining more and more attention, so research in the context of identification of EOs commissions and potential methods of its use is of high significance. The general idea of this work remains unclear to me, hence my decision to reconsider this work after major revision.

The greatest value of this work is the very detailed and accurate identification of the chemical composition of Syzygiella rubricaulis (Nees) Sthephi belonging to bryophyte species. This issue is highlighted by authors in the discussion section. In lines 94-102, authors indicated that apart from exploring chemical composition of this EOs, the aim of this work was to investigate its antimicrobial and antioxidant activity. Moreover, authors indicated that “research represent crucial step in advancing the understanding of essential oils in mosses, potentially paving the way for the development of new pharmaceutical” and that “This research holds promise for both scientific advancements and the commercial utilization of bryophytes in Ecuador”. In my opinion, it is a positive thing that the performed research at least partially contributes to the creation of knowledge that can be commercially used. However, taking into consideration the results presented by the authors, this is too far-reaching vision of the use of these results. The vision is so distant that I consider it inappropriate to point it out in this scientific article.

Answer: Thank you for your suggestion.

A more conservative statement was placed instead. Line 96.

“The research represents a significant step in advancing the understanding of essential oils in bryophytes. However, it is important to note that the investigation is still in its early stages and further studies and validations are needed before drawing conclusive implications for future research development in the pharmaceutical industry”.

Suggestion.

Authors present results for: (1) MIC values for selected gram-positive and gram-negative bacteria as well as yeasts and sporulated fungi; (2) Scavenging radical capacity and (3) AChE Analysis. All these results are presented with a reference to commercially available products there were used as a positive control.

As for the results for MIC, it is shown that S.rubricaulis EOs have activity against the Gram-positive bacteria E. faecium ATCC® 27270 and L. monocytogenes ATCC® 19115 with a MIC value of 500 μg/mL. Authors indicated in line 131, that EO demonstrated its highest inhibitory potential against indicated microbes. It is an exaggerated statement, especially that there are no other EOs from different bryophyte species that were subjected to study. Based on presented results only, there is no commercial potential for this EOs, as the difference between commercial products and tested EOs is far too big. Presented results indicate that this EOs have such activity, but the MIC values cannot be compared to anything (meaning other EOs or other products different than used reference), and thus cannot be further discussed.  The same applies for other results presented in this paper.

Answer. Thank you for your observation.

The statement was corrected according to the observation. In lines 242 to 341, and lines 358 to 366 there is a comparison of the antimicrobial activity of different related species and we added a more detailed discussion, however, there is no extensive literature information to compare about the antimicrobial potential of essential oils from mosses.

Suggestion.

As for the results for Scavenging radical capacity, there is a different value for DPPH assay provided in the text (line 142), which is equal to 2650.23 and in the table 3, which is equal to 1313.73.

Answer. Thank you for your observation.

Data in table 3 has been corrected according to the information stated in the main text.

Suggestion.

The results obtained for EOs test are closest to the reference value for the AChE study. IC50 value for this assay for EOs is equal to 26.75 μg/mL and the value for synthetic drug donepezil hydrochloride, commonly used for treating mild to moderate forms of Alzheimer's dementia is equal to 12.40 μg/mL.

However, in this case we rely only on numerical values and we do not know anything about other products or active substances showing such activity (both in terms of other EOs and other commercially available products). If the authors wanted to base their work on the value resulting from this result, it would be necessary to present the relationship between the tested activity and Alzheimer's disease.

Answer. Thank you for your observation.

Units for donepezil were corrected to nM. It was a mistake. Donepezil it is used as a positive control due to its strong activity in the order of nanomolar against AChE, in vitro, and it was not used to compare it against the activity of the EO. It is used as an internal control to validate the assay only.

Reviewer observation

The authors identified a given activities, but its identification alone is not enough for such far-reaching conclusions about a groundbreaking study (line 94). Especially since, as the authors point out (e.g. Lines 242-244, lines 230-235), it was suspected that this type of EOs (meaning EOs obtained from S. rubricaulis or from bryophytes species) would contain compounds with such activities. Thus, this result should be treated more like a confirmation rather than discovery of such activity. In my opinion, the work should be supplemented with the results for other EOs of S. rubricaulis or bryophytes origin. Different activities of tested EOs could then be correlated with different EOs composition. Such discussion would be of scientific value. And only after such research and discussion, it would be possible to come to the conclusion that given EOs has the highest activity among testes EOs, which provides grounds for conducting further research in this area. Following the authors way of thinking presented in this paper, each identification of a given activity in each of the EOs provide grounds for commercial application, which is not entirely true.

Current form of the paper makes the discussion of the results pointless, which is the main drawback of this work. If the authors have any results for other oils, or will conduct additional analyzes within a reasonable time, I believe that the work will have the potential to be published Plants Journal.

Answer to reviewer

Thank you for your great suggestions.

As pointed in the discussion section, the antimicrobial activity exerted by the EO displayed a moderate effect and we recommended additionally to study the non-volatile fraction, such an extract, like other authors did. Additionally, we made a better discussion including literature information of the chemical composition of others bryophytes, chemical composition and biological activities, in order to compare our results and give a better correspondence with our conclusions. All the corrections have been made in track change mode since line 258

Reviewer 2 Report

Comments and Suggestions for Authors

Since I am a microbiologist, my suggestions are directed to this aspect of this study.

Many names of microorganisms (but also plants) are not italics which is not appropriate for professionals and scientists such as the Authors of this paper. Please, revise this immediately. 

The obtained MIC values are very high, I am not sure that we can speak about real biocide affect and some applicability of these concentrations for further (industrial) application. Please, expand the discussion about obtained values for this view. Also, expand comparison with the latest literature.

I strongly suggest that cocci bacteria do not select in gram-positive bacteria, since this kind of division is directed to rod-shaped bacteria. Please, revise.

Campylobacter is a microaerophilic bacterium, not anaerobic. Please, revise.

Explanation of the selection of antibiotics as control and interaction with plant samples is required.

If you have the opportunity to do additional tests from the aspect of antimicrobial testing, it will be upgraded in this paper. If not, you cannot speak about antimicrobial activity, only about antimicrobial potential. So, revise it in the direction for a better understanding of this DIFFERENCE.

Selection of microorganisms for study and their source and origin must be added.

Comments on the Quality of English Language

Some basic changes can improve the quality of the paper.

Author Response

March, 2024

Prof. Dr. Dilantha Fernando

Editor in Chief

Plants

Subject: Online Manuscript Submission

To Editor

    I am submitting a required revision for consideration in Plants journal with the changes made in the manuscript with the track-change mode, for all the suggestions made by reviewers as described below:

 REVIEWER 2

Thank you for your valuable feedback, below you can find our responses.

Since I am a microbiologist, my suggestions are directed to this aspect of this study.

Suggestion.

Many names of microorganisms (but also plants) are not italics which is not appropriate for professionals and scientists such as the Authors of this paper. Please, revise this immediately. 

Answer to reviewer

Thanks for the observations. All the names of species have been corrected accordingly.

Suggestion

The obtained MIC values are very high, I am not sure that we can speak about real biocide affect and some applicability of these concentrations for further (industrial) application. Please, expand the discussion about obtained values for this view. Also, expand comparison with the latest literature.

Answer to reviewer

Thank you for your observation. We made an extensive comparison of our data with literature and all the details are in the discussion section according to your feedback.

Suggestion

I strongly suggest that cocci bacteria do not select in gram-positive bacteria, since this kind of division is directed to rod-shaped bacteria. Please, revise.

Answer to reviewer

Thank you for your observation. It was corrected accordingly.

Suggestion

Campylobacter is a microaerophilic bacterium, not anaerobic. Please, revise.

Answer to reviewer

Terminology has been corrected accordingly.

Suggestion

Explanation of the selection of antibiotics as control and interaction with plant samples is required.

Answer to reviewer

Thank you for your observation.

Broad-spectrum antibiotic and antifungal agents were chosen as internal controls for the test. All the strains were found to be sensitive to these antimicrobial agents, which are currently in use in our laboratory and have been previously documented for their activity in several published manuscripts. No interaction with the essential oil was observed, as the controls were tested separately

Suggestion

If you have the opportunity to do additional tests from the aspect of antimicrobial testing, it will be upgraded in this paper. If not, you cannot speak about antimicrobial activity, only about antimicrobial potential. So, revise it in the direction for a better understanding of this DIFFERENCE.

Answer to reviewer

Thank you for your observation. It was corrected accordingly.

Selection of microorganisms for study and their source and origin must be added.

Reviewer 3 Report

Comments and Suggestions for Authors

A brief summary 

The authors research focuses on exploring the chemical composition and some biological properties of the essential oil derived from Syzygiella rubricaulis (Nees) Stephani, a bryophyte species. GC-MS and GC-FID were used to qualitative and quantitative study.  Then the authors reported inhibitory effect against acetylcholinesterase (AChE) with an IC50 value of 26.75±1.03 μg/ml and a moderate antimicrobial (MIC 500 μg/mL, Enterococcus faecium, Lysteria monocytogenes) and antioxidant effect (ABTS: SC50 343.38 and DPPH 2650.23 μg/mL).

General concept comments
The study is new and original. The manuscript is clear and well presented. The references are recent and inherent, but not correctly reported as year should be in bolt and DOI should be correctly reported. Please check all the references. Results are reproducible and the design appropriate. The conclusions are consistent with the evidence and arguments presented. The ethics statements and data availability not applicable.

Specific comments

Figure 2 should be replaced with a more solved one. Table 2 is badly structured and not clear. I suggest to replace it or to create a diagram.

“In vitro” and “Syzygiella rubricaulis” in italics in the whole manuscript.

Author Response

REVIEWER 3

Thank you for your valuable feedback, below you can find our responses.

A brief summary 

The authors research focuses on exploring the chemical composition and some biological properties of the essential oil derived from Syzygiella rubricaulis (Nees) Stephani, a bryophyte species. GC-MS and GC-FID were used to qualitative and quantitative study.  Then the authors reported inhibitory effect against acetylcholinesterase (AChE) with an IC50 value of 26.75±1.03 μg/ml and a moderate antimicrobial (MIC 500 μg/mL, Enterococcus faecium, Lysteria monocytogenes) and antioxidant effect (ABTS: SC50 343.38 and DPPH 2650.23 μg/mL).

General concept comments
The study is new and original. The manuscript is clear and well presented. The references are recent and inherent, but not correctly reported as year should be in bolt and DOI should be correctly reported. Please check all the references. Results are reproducible and the design appropriate. The conclusions are consistent with the evidence and arguments presented. The ethics statements and data availability not applicable.

Specific comments

Suggestion

Figure 2 should be replaced with a more solved one. Table 2 is badly structured and not clear. I suggest to replace it or to create a diagram.

Answer to reviewer

Thank you for your suggestion. It has been addressed properly.

Suggestion

“In vitro” and “Syzygiella rubricaulis” in italics in the whole manuscript

Answer to reviewer

Thank you for your suggestion. It has been addressed properly.

All the authors agreed to the final version of this manuscript. We all thank to the reviewers and editor for the opportunity to publish our job in such an important and prestigious journal.

Round 2

Reviewer 1 Report

Comments and Suggestions for Authors

Changes made in the text of the work are appropriate. The narrative of the work has changed and is now focused on pilot research aimed at setting a direction or providing a basis for further research, rather than presenting very groundbreaking research with high application potential. Current way of presenting the meaning and significance of these results definitely suits them better.

I understand the need to publish pilot results in order to continue research in this area. I strongly believe that the authors will not abandon this subject of research and will continue research on the biological activities of this EO and that in subsequent works they present a comparison of the activities of various EOs originating from Bryophyte species.

Reviewer 2 Report

Comments and Suggestions for Authors

The authors have implemented all suggestions in the new version of the manuscript, therefore I recommend to publish this paper in present form.